# Rationale and Design of the Future Optimal Research and Care Evaluation in Patients with Acute Coronary Syndrome (FORCE-ACS) Registry: Towards “Personalized Medicine” in Daily Clinical Practice

**DOI:** 10.3390/jcm9103173

**Published:** 2020-09-30

**Authors:** Dean R. P. P. Chan Pin Yin, Gert-Jan A. Vos, Niels M. R. van der Sangen, Ronald Walhout, R. Melvyn Tjon Joe Gin, Deborah M. Nicastia, Jorina Langerveld, Daniël M. F. Claassens, Marieke E. Gimbel, Jaouad Azzahhafi, Willem L. Bor, Tom Oirbans, Johan Dekker, Georgios J. Vlachojannis, Rutger J. van Bommel, Yolande Appelman, José P. S. Henriques, Wouter J. Kikkert, Jurriën M. ten Berg

**Affiliations:** 1Department of Cardiology, St. Antonius Hospital, 3435 CM Nieuwegein, The Netherlands; d.claassens@antoniusziekenhuis.nl (D.M.F.C.); m.gimbel@antoniusziekenhuis.nl (M.E.G.); j.azzahhafi@antoniusziekenhuis.nl (J.A.); w.bor@antoniusziekenhuis.nl (W.L.B.); t.oirbans@antoniusziekenhuis.nl (T.O.); johan.dekker@antoniusziekenhuis.nl (J.D.); j.ten.berg@antoniusziekenhuis.nl (J.M.t.B.); 2Department of Cardiology, Amsterdam UMC, University of Amsterdam, Amsterdam Cardiovascular Sciences, 1105 AZ Amsterdam, The Netherlands; n.m.r.vandersangen@amsterdamumc.nl (N.M.R.v.d.S.); j.p.henriques@amsterdamumc.nl (J.P.S.H.); 3Department of Cardiology, Hospital Gelderse Vallei, 6716 RP Ede, The Netherlands; WalhoutR@zgv.nl; 4Department of Cardiology, Rijnstate Hospital, 6815 AD Arnhem, The Netherlands; mtjon@rijnstate.nl; 5Department of Cardiology, Gelre Hospitals, 7334 DZ Apeldoorn, The Netherlands; d.nicastia@gelre.nl; 6Department of Cardiology, Rivierenland Hospital, 4002 WP Tiel, The Netherlands; Jorina.Langerveld@zrt.nl; 7Department of Cardiology, University Medical Center Utrecht, 3584 CX Utrecht, The Netherlands; G.Vlachojannis@umcutrecht.nl; 8Department of Cardiology, Tergooi Hospital, 1261 AN Blaricum, The Netherlands; rvanbommel@tergooi.nl; 9Department of Cardiology, Amsterdam UMC, VU University, Amsterdam Cardiovascular Sciences, 1081 HV Amsterdam, The Netherlands; y.appelman@amsterdamumc.nl; 10Department of Cardiology, Onze Lieve Vrouwe Gasthuis, 1091 AC Amsterdam, The Netherlands; w.j.kikkert@amsterdamumc.nl

**Keywords:** acute coronary syndrome, antiplatelet therapy, multicenter registry

## Abstract

Diagnostic and treatment strategies for acute coronary syndrome have improved dramatically over the past few decades, but mortality and recurrent myocardial infarction rates remain high. An aging population with increasing co-morbidities heralds new clinical challenges. Therefore, in order to evaluate and improve current treatment strategies, detailed information on clinical presentation, treatment and follow-up in real-world patients is needed. The Future Optimal Research and Care Evaluation in patients with Acute Coronary Syndrome (FORCE-ACS) registry (ClinicalTrials.gov Identifier: NCT03823547) is a multi-center, prospective real-world registry of patients admitted with (suspected) acute coronary syndrome. Both non-interventional and interventional cardiac centers in different regions of the Netherlands are currently participating. Patients are treated according to local protocols, enabling the evaluation of different diagnostic and treatment strategies used in daily practice. Data collection is performed using electronic medical records and quality-of-life questionnaires, which are sent 1, 12, 24 and 36 months after initial admission. Major end points are all-cause mortality, myocardial infarction, stent thrombosis, stroke, revascularization and all bleeding requiring medical attention. Invasive therapy, antithrombotic therapy including patient-tailored strategies, such as the use of risk scores, pharmacogenetic guided antiplatelet therapy and patient reported outcome measures are monitored. The FORCE-ACS registry provides insight into numerous aspects of the (quality of) care for acute coronary syndrome patients.

## 1. Introduction

Over the past decades, the overall mortality of patients admitted with acute coronary syndrome (ACS) has decreased [1,2]. Simultaneously, the implementation of evidence-based treatments in ACS patients has led to a lower risk of recurrent ischemic events [1,2]. However, nationwide registries in different countries have demonstrated that approximately 20–30% of patients have a recurrent cardiovascular event (i.e., myocardial infarction (MI), stroke or cardiovascular death) in the five years following their initial admission [3,4]. For example, a conjoined registry of ACS patients (*n* = 3721) from the United Kingdom and Belgium has shown that one in five ACS patients are deceased within five years, implying that careful monitoring beyond the first year and additional secondary prevention measures are warranted [3].

Accordingly, several steps have been taken to improve secondary prevention. During the past 20 years, the introduction of more potent antithrombotic treatment has been associated with a substantial reduction of ischemic events and improved survival, but at the cost of an increase in bleeding events [5]. More recently, new patient-tailored Dual AntiPlatelet Therapy (DAPT) strategies have been developed, such as the use of risk scores to guide DAPT duration after ACS, potentially allowing for a better assessment of bleeding and thrombotic risk in individual patients [6]. For example, the PREdicting bleeding Complications In patients undergoing Stent implantation and subsEquent Dual AntiPlatelet Therapy (PRECISE-DAPT) score and the DAPT score are promising tools for personalized decision-making regarding the optimal duration of DAPT [7,8]. However, a necessary prospective validation of strategies incorporating these scores has not yet been performed. In addition, preventive strategies have often arisen from large randomized controlled trials (RCTs). While RCTs have a high internal validity, they might not reflect the patient population in daily clinical practice. For example, women, older patients and patients with a bleeding history are underrepresented in most RCTs, possibly influencing the value of the outcome of RCTs for the elderly [9]. In contrast to most RCTs, registries have less strict inclusion criteria (e.g., no age limit and no limitations on comorbidities) and therefore better reflect the real-world population [10,11,12,13,14].

Furthermore, different regional treatment protocols co-exist for patients with ACS, leading to a wide variety in treatment between hospitals. Currently ongoing nationwide registries gain insight into cardiac treatment strategies and intend to improve the quality of care, but are often limited, using minimal datasets for data collection where important details on treatment are lacking, such as antithrombotic treatment strategy (type, duration and compliance) and both ischemic and bleeding end points during follow-up. In this manuscript, we describe how the FORCE-ACS (Future Optimal Research and Care Evaluation: Towards “personalized medicine” in patients with Acute Coronary Syndrome in daily clinical practice) registry allows the monitoring of ACS patients across the Netherlands and overcomes some of the limitations of previous registries. To illustrate the scope of research projects conducted within the registry, we will elaborate on the study design of the project, evaluating the effects of risk score-guided DAPT duration.

## 2. Methods

### 2.1. Study Objectives

The primary objectives of the FORCE-ACS registry are: (1) to create an ongoing registry of patient characteristics and treatment strategies, therefore facilitating future research, enabling personalized therapy and aiming to improve quality of care for patients with ACS; (2) to assess the efficacy and safety of new drugs or strategies versus the current treatment and, if available, to compare the results of these novel treatments in daily practice to RCTs; (3) to evaluate and improve current health care pathways between different Dutch hospitals (e.g., the collaboration between interventional and non-interventional cardiac centres), which can also be used for value-based health-care initiatives; and (4) to increase the adherence to current guidelines to improve secondary prevention. All data needed for mandatory quality-of-care registries are also collected in the FORCE-ACS registry.

Research questions and hypotheses with a significant focus on the secondary prevention of recurrent ischemic events are proposed. A major topic of interest is the individualization of DAPT duration based on a patient’s ischemic and haemorrhagic risk. We hypothesize that a personalized DAPT duration in patients with ACS, using the PRECISE-DAPT score and DAPT score, is superior to the current standard treatment with regards to ischemic and bleeding outcomes (see Box 1). Other research topics that will be investigated in this registry include the efficacy and safety of pharmacogenetic guided treatment for antiplatelet therapy, dual versus triple therapy in patients with an indication of oral anticoagulation, the evaluation of optimal cardiovascular risk management (in important subgroups such as the elderly and women) and patient-related outcome measures (PROMs) in patients ≥ 1 year after ACS. End points, sample sizes and statistical analyses are formed per hypothesis. Future research questions and hypotheses may be proposed as the registry progresses. All research questions and hypotheses are formulated in the research protocol (Clinicaltrials.gov; NCT03823547).

### 2.2. Study Design and Population

The FORCE-ACS registry (ClinicalTrials.gov Identifier: NCT03823547) is a multicenter, prospective registry including patients from nine non-interventional and interventional cardiac centers located in different regions of the Netherlands. The registry is coordinated by investigators at St. Antonius Hospital (Nieuwegein, The Netherlands) and the Amsterdam University Medical Center, location AMC (Amsterdam, The Netherlands). An overview of the participating institutions is provided in Table 1 and their geographic locations are shown in Figure 1. Since January 2015, consecutive adult patients (18 years or older) admitted with suspected ACS, i.e., unstable angina or non-ST-elevation myocardial infarction (NSTEMI) and ST-elevation myocardial infarction (STEMI), are prospectively included in this registry. Before May 2018, there was no legal obligation to obtain written informed consent for registries that were not subjected to the Medical Research Involving Human Subjects Act in the Netherlands. Since the enforcement of the General Data Protection Regulation (GDPR), all patients included in the registry have provided written informed consent. Patients who were included before May 2018 were contacted and asked to provide written informed consent. Patients who deceased during admission before informed consent was obtained (e.g., cardiac arrest due to ACS) will be included pseudonymously, unless objections could have been reasonably presumed. It will be encouraged to follow the current guidelines of the European Society of Cardiology (ESC) with final decisions at the discretion of the treating physician [6,15,16,17,18].

### 2.3. Data Collection

Data will be prospectively collected by members of the investigation team during admission and after 1, 12, 24 and 36 months(s) as illustrated in Figure 2. Automated data extraction from electronic medical records (EMR) will be applied as much as possible to facilitate the collection of significant amounts of data. Data extraction will be performed periodically (e.g., monthly) or, if possible, continuously from EMRs according to standardized operating procedures (SOPs). Data will be stored anonymously in a specially designed database and the input of data will be performed using a special front-end interface (REDCap, Vanderbilt University) [19]. All primary and secondary end points during an at least three-year follow-up will be recorded by the investigators, using hospital EMRs, general practitioner records, pharmacy records and patient questionnaires. Patients will be contacted at 1, 12, 24 and 36 months after discharge to ask whether they have been admitted to other hospitals and to assess their quality of life using the Short Form 12 (SF-12) and/or Seattle Angina Questionnaires [20,21]. Additionally, patients will be asked to keep track of any changes in antiplatelet regimen (e.g., temporary cessation of aspirin or P2Y_12_-inhibitor) and to provide a list of all medications they use. If necessary, this data will be supplemented with information from pharmacies.

### 2.4. Study End Points

Major end points include all-cause mortality, stroke, MI, hospital admission for ACS, stent thrombosis (classified according to the Academic Research Consortium criteria), revascularization and all bleeding requiring medical intervention, defined using different bleeding classifications (Bleeding Academic Research Consortium (BARC), Thrombolysis in Myocardial Infarction (TIMI), Global Utilization Of Streptokinase And Tpa For Occluded Arteries (GUSTO) and Platelet Inhibition and Patient Outcomes (PLATO)) [22,23]. The following end points will also be collected: mortality classification into cause of death (i.e., cardiovascular or non-cardiovascular cause of death), (temporary) cessation and duration of antiplatelet therapy after ACS and PROMs as reported in quality of life questionnaires (e.g., SF-12 and Seattle Angina Questionnaires) (see Appendix A for a detailed description of all study end points). If feasible, the collection of additional relevant end points may follow as the study progresses.

### 2.5. Sample Size and Statistical Analysis

This ongoing registry is not limited to a certain size; sample sizes are calculated based on individual study hypotheses. Categorical data will be presented as number of patients and percentages. Continuous data will be presented as mean ± standard deviations, median (25th–75th percentile) or range, as appropriate. Missing data are handled on an analysis-specific basis. Event rates are estimated using the Kaplan–Meier method, and Cox proportional hazard models will be used to estimate adjusted between-group effects. Other analytical statistics are used according to individual research questions.

### 2.6. Quality Control

All members of the investigation team are trained for adequate data entry with an SOP guide. Around 5–10% (randomly selected) of all baseline and clinical data and 100% of all diagnoses at discharge will be reviewed by the coordinating investigators. All major end points will be reviewed by a clinical event committee, consisting of at least two independent investigators. Furthermore, the web-based REDCap database automatically provides periodic data checks and queries for inconsistent or out-of-range data in order to urge the user to review and correct the data if needed.

### 2.7. Regulation Statement and Informed Consent

The medical ethics review committees of the various participating centers have confirmed that the Medical Research Involving Human Subjects Act does not apply to this registry. The FORCE-ACS registry will be conducted according to the Medical Treatment Agreement Act, the General Data Protection Regulation and the Declaration of Helsinki. All patients will have to provide written informed consent for participation in the registry and will have the possibility to discontinue participation at any time.

### 2.8. The FORCE-ACS Database Framework

The FORCE-ACS registry framework will enable the investigation of multiple research questions using a multitude of study designs. The registry can be used for prospective cohort studies, in which the effect of different treatment strategies may be compared and evaluated over time. Propensity matched analyses will be applied if suitable. In addition, retrospective (post-hoc) analyses can be conducted at any moment in time, depending on the availability of data for a specific research question. This will also provide valuable data for case-control studies in the future (e.g., for analysis of relatively rare complications such as stent thrombosis). The FORCE-ACS framework provides a unique foundation for multicenter registry-based randomized controlled trials (r-RCT). Since full data collection and follow-up are already performed, a randomization module can be added to enable pragmatic clinical trials. Additional data elements or biomedical data (i.e., genetic data) can be added for selected patient groups.

Box 1Evaluation of risk score-guided Dual AntiPlatelet Therapy (DAPT) duration within the Future Optimal Research and Care Evaluation in patients with Acute Coronary Syndrome (FORCE-ACS) registry.
**Risk score-guided DAPT duration**
Within the FORCE-ACS registry, a prospective, multicenter cohort study is currently being performed. Up to July 2018, DAPT duration was not determined using risk scores in any of the participating hospitals. Most patients received 12 months DAPT, which was only shortened or prolonged based on clinical judgement. From July 2018, a risk score-guided DAPT duration using the PRECISE-DAPT score and the DAPT score was encouraged in all participating centers. The cohort is divided into two groups:
Standard treatment group: DAPT duration not based on risk scoresRisk score-guided group: DAPT duration based on risk scores using the following treatment scheme (see Figure 3)ACS patients with a high bleeding risk (PRECISE-DAPT score ≥ 25) will receive a short DAPT duration (6 months); patients without high bleeding risk (PRECISE-DAPT score < 25) will receive at least 12 months of DAPT. In patients who have endured one year of DAPT without any events, patients with high ischemic risk (DAPT score ≥ 2) will receive an extended DAPT duration (at least 30 months post-ACS); in patients without a high ischemic risk (DAPT score < 2), DAPT will be discontinued after 12 months. All patients enrolled in the FORCE-ACS registry and with an indication for DAPT at discharge will be eligible for this analysis. Patients with a concomitant use of oral anticoagulants, a contraindication for aspirin or a P2Y_12_-inhibitor or implementation of a bio-absorbable vascular stent will be excluded from analysis. The primary end point for this study is defined as the composite of all-cause mortality, MI, stent thrombosis, stroke and BARC 3 or 5 bleeding at a 24 months follow-up. The expected event rate of the primary end point was estimated at 15% at a 24 months follow-up in the standard treatment group. Furthermore, we expect to enroll patients in a 3 (standard treatment) to 1 (risk score-guided treatment) ratio. Using an α of 0.05 and a 1-β of 0.80, we would need a study population of 5554 to show a relative reduction of 20% at the primary end point. We expect that approximately 10% of patients without a final diagnosis of ACS and 15% of patients with an indication for oral anticoagulation will be excluded from analysis. If the loss to follow-up is limited to 5% after 24 months, a total study population of approximately 8000 ACS patients is required. Primary analysis will compare the treatment groups from the time of inclusion to the first occurrence of any event in the composite end point using Kaplan–Meier cumulative event-free curves and formal statistical testing with the log-rank test. For measure of the treatment effect, hazard ratios will be computed with a Cox proportional hazard regression.

**Figure 3 jcm-09-03173-f003:**
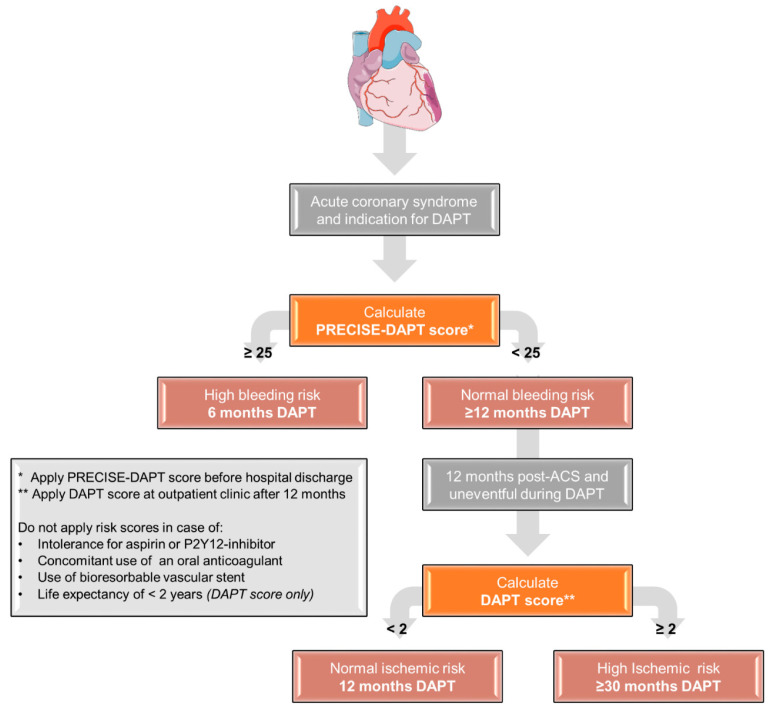
Flowchart illustrating decision-making regarding the optimal treatment duration with dual antiplatelet therapy using risk scores. Abbreviations: ACS, acute coronary syndrome; DAPT, Dual AntiPlatelet Therapy.

## 3. Discussion

In order to evaluate and improve the current care of ACS patients, detailed information regarding clinical presentation, treatment and follow-up is needed. While there are ongoing national and international registries, they often do not include the full spectrum of ACS patients, collect a limited scope of data or have a short follow-up period (i.e., 1 year or less).

In the Netherlands, ongoing nationwide registries, such as the Netherlands Heart Registration (NHR), are mainly focused on cardiac interventions (e.g., percutaneous coronary interventions (PCI) or coronary artery bypass grafting) and are limited in scope. Detailed information on important outcomes (e.g., bleeding) is usually not registered or is scored inconsistently. In addition, medication adherence is not registered and there is usually no active follow-up (e.g., with questionnaires).

Though most patients admitted with ACS will undergo PCI, a significant number of patients are also treated conservatively. Most often, a conservative strategy is pursued in fragile, older patients with multiple comorbidities. They represent a growing group of patients that have traditionally been unrepresented in RCTs, despite the complexity physicians encounter when treating them. The same is true for women, who are less likely to have obstructive coronary artery disease and might benefit from alternative treatment strategies as compared to men [24]. Registration of important clinical parameters and outcome data will provide clinical guidance for optimal treatment in these patients.

Feedback from this detailed dataset and long-term outcome data will be used to improve adherence to the current guidelines and establish uniform treatment in participating hospitals. Relevant and unambiguous end point definitions and scoring end points from patient files and pharmacy records are an integral part of the registry, and well conducted follow-up to minimize loss of valuable data is crucial in this registry.

### Future Perspective

Most hospitals are converting to next generation EMRs, enabling the standardized registration of baseline data and scoring of outcomes, which will make more efficient and complete data registration possible. Export modules to connect various commercial EPDs to the FORCE-ACS framework in REDCap are being developed. A centralized database enables cooperation for the follow-up of outcomes during the outpatient setting in referring clinics after PCI. Quality of care measures and peer comparison in the region will be possible on an automated basis (i.e., a quarterly report). In addition, the FORCE-ACS registry can serve as a framework and infrastructure for multi-center prospective studies, which is especially useful for post-marketing surveillance (phase 4) studies. This framework would make research more cost-effective and would speed up implementation.

Medical knowledge is advancing at an ever-increasing rate, and clinicians face the challenge of keeping up with all these new advancements. Excessive demands can lead to suboptimal guideline adherence and variation among hospitals [25]. Clinical decision support systems can help clinicians decide on optimal therapies for their patients on a more personalized basis, incorporating clinical parameters and even genetic data. Deep learning based on neural networks will provide new possibilities to identify patterns in clinical data and provide risk assessments for adverse outcomes. These systems need to be developed, evaluated and improved. Furthermore, PROMs will have an increasingly important role in guiding treatment. Our database will enable the investigation of initiatives to improve treatment and guideline adherence in a continuous feedback loop.

The FORCE-ACS registry is supported by the Hart Beter Society (regional collaboration of cardiologists). Further collaboration with hospitals, ambulance organizations and general practitioners will be sought in other regions across the Netherlands.

## 4. Conclusions

The multicenter, prospective registry FORCE-ACS is an open end large-scale real-world registry of patients with ACS, used to evaluate the clinical long-term impact of medication, devices and diagnostic tools, especially with a view to personalized medicine and quality of care. This will open a new frontier in clinical research in ACS patients.

## Figures and Tables

**Figure 1 jcm-09-03173-f001:**
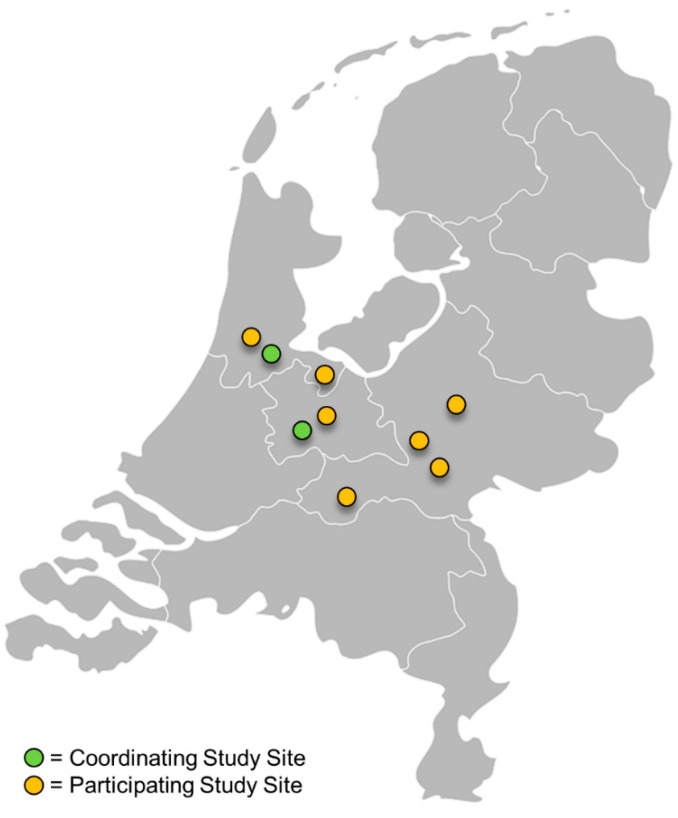
Geographical locations of cardiac centers participating in the Future Optimal Research and Care Evaluation in patients with Acute Coronary Syndrome (FORCE-ACS) registry.

**Figure 2 jcm-09-03173-f002:**
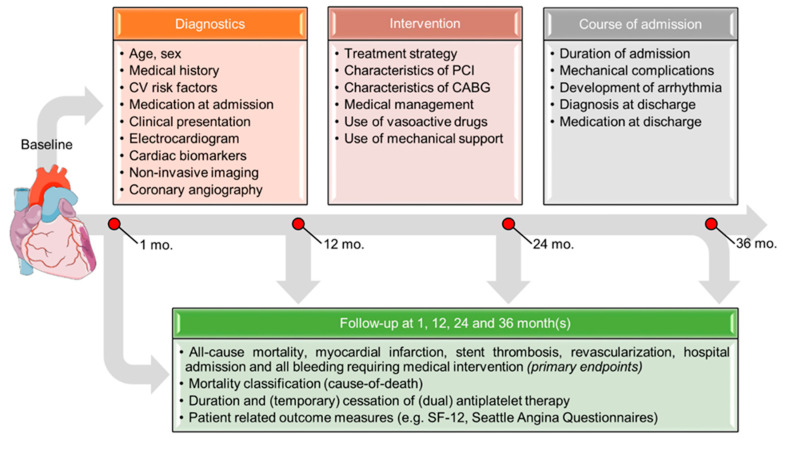
Flowchart illustrating data collection in the in Future Optimal Research and Care Evaluation in patients with Acute Coronary Syndrome (FORCE-ACS) registry. Abbreviations: CABG, coronary artery bypass grafting; CV, cardiovascular; PCI, percutaneous coronary intervention; and SF, short form.

**Table 1 jcm-09-03173-t001:** Overview of participating secondary and tertiary cardiac centers in Future Optimal Research and Care Evaluation in patients with Acute Coronary Syndrome (FORCE-ACS) registry.

Hospital (Location)	Hospital Type	Start Inclusion
Coordinating study sites		
1. St. Antonius Hospital (Nieuwegein, the Netherlands)	Interventional cardiac center	2015
2. Amsterdam UMC, location AMC (Amsterdam, the Netherlands)	Interventional cardiac center	2019
3. Hospital Gelderse Vallei (Ede, the Netherlands)	Non-interventional cardiac center	2018
4. Rijnstate Hospital (Arnhem, the Netherlands)	Interventional cardiac center	2019
5. Gelre Hospitals (Apeldoorn, the Netherlands)	Non-interventional cardiac center	2018
6. Rivierenland Hospital (Tiel, the Netherlands)	Non-interventional cardiac center	2018
7. Amsterdam UMC, location VUmc (Amsterdam, the Netherlands)	Interventional cardiac center	2020
8. Tergooi Hospital (Hilversum, the Netherlands)	Interventional cardiac center	2020
9. University Medical Center Utrecht (Utrecht, the Netherlands)	Interventional cardiac center	2020

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
