# Peer review of "Rationale and Design of the Future Optimal Research and Care Evaluation in Patients with Acute Coronary Syndrome (FORCE-ACS) Registry: Towards “Personalized Medicine” in Daily Clinical Practice"

_jcm, 2020, doi:10.3390/jcm9103173_

Round 1

Reviewer 1 Report

The authors report on their design of an ACS registry that will be used for prognostic information as well as the conduction of registry-based randomized studies with a focus on DAPT duration. The manuscript is well-written and the study is well-planned. There are several issues that if addressed could improve the manuscript.

Major comments

  • “Other research topics that will be investigated in this registry include the efficacy and safety of pharmacogenetic guided treatment for antiplatelet therapy.” Are the authors going to collect blood samples from the participants in the registry? Are the biochemical markers used for prognosis including the ones used for DAPT score be performed in a single lab or in the lab of each center? If each center performs their own measurements on biochemical markers as well as to adjust for any difference in treatment of ACS adjustment of the results for each center should be performed.
  • Do the authors know how many centers will be involved and what will be their capabilities i.e. interventional vs non-interventional, university vs non-university etc.?
  • The authors report 3 figures in the manuscript but at least I cannot see them in my copy of their submission. Please include these figures in the revised submission manuscript.
  • Of extreme interest are socioeconomical data and lifestyle risk factors (such as smoking, alcohol, physical activity, sexual activity). Are these factors going to be covered by the registry and if yes how exactly are they going to be assessed?
  • Will the data be available to others or will be used only by the center that has the database?
  • Will patients enrolled in other studies (especially randomized studies) be eligible to enroll?
  • Please cite the references you used to estimate the sample size for the event rate as well as the 3 to 1 distribution of standard treatment compared to algorithm-guided treatment?
  • Are there any other exams not clinically indicated expected to be performed in these patients i.e. MRI, CTCA etc.?

Reviewer 2 Report

This multicenter, prospective registry represents a  large-scale real-world registry 230 of patients with ACS, used to evaluate the clinical long-term impact of medication, devices and 231 diagnostic tools. Despite the relatively small sample size, generalization of outcomes can be helpful to capture real world data in such community . DAPT is a big deal in ACS population and the collected information if collected in depth will be helpful to enlighten cardiologist and non cardiologist for DAPT regimens and special population as CKD, DM, Afib ..etc. 

Reviewer 3 Report

Dear editor and Authors. 

I wish to thank for the opertunity to review the current manuscript.

I would like to acknowledge the authors and registry coordinators for taking on the task to develop a detailed registry of al-comers with a suspected AMI. The registry seem to be of high quality and contain very relevant data.

I find it is a bit difficult to thoroughly review the current manuscript as it is not a study. Rather it is just a detailed description of the planned framework for the FORCE-ACS registry. Thus - I find it is difficult to comment specifically on the design of the registry. 

The manuscript is easy to read and the english language is very good.

I have the following comments and questions that I do find need additional answering.

Study population: "Suspected AMI". How is this defined? This is indeed a very heterogeneous group. The usual estimate is that only 1-2 of 10 with a suspected AMI actually do have the final diagnosis. Will the registry really contain data of 5-10 times as many patients than those who actually end up having an ACS diagnosis?

Type of registry: I struggle to understand the characteristics of the registry. Is it a "quality monitoring registry", or is it a "research registry"? I find this is important to state. I get the impression that the registry is primarily a "Research registry". If this is the case, then I would expect specific research questions be associated with the collected variables. To me it seems as if the authors primarily collect the data. Then later, the research questions are made. I think this should be commented on. Certainly - in the current GDPR regulations, just collecting data with no specific purpose is not necessarily justifiable.

Could the authors please comment on if there is also a quality assessment registry in the referenced centers. If this is not the case, then why not primarily design a quality assessment registry with the possibility to review guideline advised quality indicators. If additional data are needed then this could be added ad hoc. In my opinion, with such a large number of patients covered by the registry, a quality assessment registry would have priority over a research registry.

The authors mention a very high volume of data to be collected with individual validation of data and endpoints by cardiologists. Data are to be extracted from patients files using extraction algorithms etc. I am amazed if it is really possible to collect such a wast amount of data. Does the authors have any experience with the completeness of data already now? If not - when does the authors expect to have full disclosure to all the mentioned outcomes and variables.

Round 2

Reviewer 1 Report

The authors have adequately responded to the Reviewer's comments.